# Study of the Microstructure and Ring Element Segregation Zone of Spray Deposited SiC_p_/7055Al

**DOI:** 10.3390/ma12081299

**Published:** 2019-04-20

**Authors:** Hao Yang, Xin-wei She, Bin-bin Tang, Chun-mei Li, Xian-quan Jiang

**Affiliations:** 1Faculty of Materials and Energy, Southwest University, Chongqing 400715, China; yanghaolouis@163.com (H.Y.); 13983073429@163.com (X.-w.S.); 2013990014@qhu.edu.cn (B.-b.T.); lcm1998@swu.edu.cn (C.-m.L.); 2Chongqing Advanced Materials Research Center, Chongqing Academy of Science and Technology, Chongqing 401123, China

**Keywords:** spray deposition, 7055 aluminum alloy, SiC particles, diffusion, element segregation

## Abstract

Composites of 7055 aluminum (Al) matrix reinforced with SiC particles were prepared using the spray deposition method. The volume fraction of the phase reinforced with SiC particles was 17%. The effect of the introduction of SiC particles on the deposited microstructure and properties of the composites was studied in order to facilitate the follow-up study. The structure and element enrichment zone of spray-deposited SiC_p_/7055 Al matrix composites were studied by Optical Microscope (OM), X-ray diffraction (XRD), Scanning Electronic Microscopy (SEM) and Transmission electron microscopy (TEM). The results show that the reinforcement phases of the SiC particles were uniformly distributed on the macro and micro levels, and a few SiC particles were segregated into annular closed regions. C and Si on the surface of SiC particles diffused to the Al matrix. The distribution of the two elements was gradient weakening with SiC particles as the center, and the enrichment zones of Si, Mg and Cu formed in the middle of the closed annular area of a few SiC particles. The enrichment zones were mainly composed of alpha-Al, SiC, Al_2_CuMg, Al_2_Cu and MgZn_2_. AlCu and AlMgCu phase precipitate on the surface of the SiC particles, beside the particle boundary, and had the characteristics of preferred nucleation. They tended to grow at the edges and corners of SiC particles. It was observed that the formation of nanoparticles in the alloy had a pinning effect on dislocations. The different cooling rates of the SiC particles and the Al matrix led to different aluminum liquid particle sizes, ranging from 20 to 150 μm. In the region surrounded by SiC particles, the phenomenon of large particles extruding small particles was widespread. Tearing edges and cracks continued to propagate around the SiC particles, increasing their propagation journey and delaying the fracture of the materials.

## 1. Introduction

As-cast materials with a uniform composition, fine structure and high density can be obtained by spray deposition technology [1,2]. This technology has been widely used in the development of and research into aerospace alloys and high-performance military materials [3,4,5,6,7,8]. SiC particle reinforcement has the advantages of a simple process and low production cost. The reinforced material has the characteristics of high specific strength, high specific stiffness and a good thermal expansion coefficient [9,10,11,12,13,14,15,16]. Scholars at home and abroad have made high-strength aluminum–gold materials or SiC_p_/Al metal composites by spray deposition or other methods, and have studied how to use them as lightweight structural materials in weapons manufacturing and military radar packaging [17,18,19,20,21,22].

SiC particles, particle size, and the second phase are important factors affecting the mechanical properties of SiC_p_/7055 Al matrix composites. A large number of studies have shown that SiC particles can bear part of the load, influence the microstructure, and reduce the crack growth ability of the hard phase [23,24,25,26], but the pore between the enriched SiC particles will become the channel of crack growth [27]. The mechanical properties of spray-deposited composites decrease gradually due to the large particle size [28]. The second phase particles in composites are mainly MgZn_2_ and CuAl_2_, which can reduce the precipitation of the second phase and increase the solid solubility, and greatly improve the hardness of the composites [29]. 

At present, the preparation methods of particulate reinforced aluminum matrix composites mainly include the stirring casting method and the melt infiltration method, the advection casting method, the powder metallurgy method and the spray deposition method. Because the cooling rate of the stirring casting method and the melt infiltration method is low, the distribution of reinforcement particles is uneven, and the properties of the prepared materials are poor. The properties of composites obtained by advection casting and powder metallurgy are higher, but there are some problems such as degassing difficulty, structure coarsening and complex technology. SiC particulate-reinforced aluminum matrix composites prepared by spray deposition technology have the advantages of high Young’s modulus and good wear resistance [30]. Compared with traditional methods, spray deposition technology has the advantages of a simple process, low segregation, a high cooling rate, and a fine microstructure. It has become one of the common methods used to prepare aluminum matrix composites [31,32]. The development of spray deposition technology is restricted by the existence of micro-holes in ingots prepared by spray deposition, and the incomplete metallurgical bonding between particles and layers. Many scholars have adopted extrusion and rolling methods to improve the density and material properties of composites [33,34,35]. The research on materials with SiC particle-reinforced phases is mostly focused on the subsequent thermal deformation and treatment process, and there is less research on deposited SiC particles, aluminum liquid particles, second phase and SiC particle-collective interface.

In this study, 7055 aluminum alloy with excellent properties was combined with spray deposition technology, and a new aluminum matrix composite was prepared by adding 17% of SiC particle-reinforced phase. The enrichment zones of Mg, Si and Cu in the annular SiC particles were observed; these have not previously been reported in the literature. The formation mechanism of the enrichment zones and the effect of the SiC particles on the microstructure and properties of the deposited SiC particles were also studied. 

## 2. Experiment

SiC_p_/7055 aluminum matrix composites were prepared by spray deposition. The chemical composition of 7055 alloy is shown in Table 1. Before use, the particles were heated at 250 °C for 10 h to remove the crystalline water and adsorbents, thus reducing the agglomeration of SiC particles themselves. The spray deposition process parameters are shown in Table 2. The volume fraction of added alpha-SiC particles was 17%, and the density of the material was 92.3%. The deposited ingots with 160 mm diameter and 320 mm height were prepared by spray deposition. The samples with 140 mm diameter and 310 mm height were obtained by turning. The surface of the ingot was sampled along the radial direction as shown in Figure 1. The ZEISS Axiovert 200MAT metallographic microscope (ZEISS, Oberkochen, Germany) was used to observe the aluminum liquid particle structure and SiC particle distribution of the composites. Scanning electron microscopy was used to observe the microstructure and fracture morphology, and phase analysis was carried out by an energy dispersive spectrometer (EDS). X-ray diffraction was used to identify the type of phase and transmission electron microscopy was used to observe the pinning effect of nanoparticles on dislocations.

## 3. Results and Analysis

### 3.1. Metallographic Microstructure Analysis

As shown in Figure 1a, samples were taken at different locations and orientations of deposited SiC_p_/7055Al composites by spray deposition. At the same height from the bottom of the sample, the radial metallographic images of the center of the cross-section (Figure 1b(1)), the center of the radius of the cross-section (Figure 1b(2)), and the edge of the cross section (Figure 1b(3)) were compared. 

Figure 2 shows the metallographic structure of spray-deposited SiC_p_/7055 Al matrix composites at different locations. From Figure 2a–c, it can be observed that the microstructures of the composites at different positions were fine and uniform. The SiC particles were about 40–50 μm in size. They were uniformly distributed in the matrix macroscopically, segregated in the micro-region, and closed annular in a few places. In previous studies, spray-deposited SiC_p_/7055 aluminum alloy had a very similar metallographic structure at different locations at the same height. The aluminum liquid particle size was equiaxed and uniform. The main particle size was 40 ~ 50 µm [36]. Figure 2d–f shows the particle structure of the 7055 alloy at different locations after introducing SiC particles. The three metallographic structures were similar, and the aluminum liquid particle sizes were quite different, basically between 20 and 150 µm. When the SiC particles distribute evenly in the micro-segregation region, there will be fine and uniformly distributed particle distribution regions in these SiC particle segregation regions. The aluminum liquid particle size was between 10 and 50 µm, which was smaller than that of the 7055 aluminum alloy sprayed without the SiC particle reinforcement phase. In the region far away from the SiC particles, there were aluminum liquid particles of size 100–200 µm. Because of the addition of the SiC, the cooling rate of the SiC particles was relatively fast during the solidification process, and there were a large number of active sites on the surface, which can create conditions for a large amount of nucleation of aluminum liquid particles. Therefore, a large number of SiC particles first nucleated in the SiC segregation zone and at the interface between the SiC and the matrix, and the number of particles was further away from the SiC region. However, due to the limited space, the particles near the SiC particles needed space and alloy composition to grow. After nucleation, the particles inhibited each other’s particle growth. On the other hand, the SiC particles hindered the direction of the particle growth. When the particles in the SiC segregation region met SiC particles in the process of particle growth, the particle boundary ceased to migrate and the particles stopped growing in that direction in the SiC segregation region. A large number of small and uniform particles were formed near the particles. After the formation of nuclei far from the SiC particles, the size of the nuclei was larger because of the high temperature, sufficient energy, alloy composition, and growth space.

### 3.2. Microstructure Analysis

Figure 3 shows the XRD pattern of the composite. Samples were tested and analyzed on a Shimadzu XRD-6100 X-ray diffractometer (Shimadzu, Kyoto, Japan). The scanning power was 4 kW, the scanning angle was 10–90°, and the scanning speed was 2 °/min. The analysis shows that the alloys in the composites were composed of alpha-Al, SiC, Al_2_CuMg, Al_2_Cu and MgZn_2_. The reinforcing phase was hexagonal crystalline alpha-SiC, and the content of MgZn_2_ was higher in the second phase.

A JSM-6610 scanning electron microscope (JEOL, Tokyo, Japan) was used to observe the microstructure of the experiment. The scanning voltage was 20 kV. Component analysis was performed using an Oxford Energy Spectrometer (Oxford instruments, Oxford, UK). Figure 4 is a scanning electron microscopic image of the microstructures of the composites. From Figure 4a, it can be observed that the SiC was segregated locally (circles indicate the aggregation distribution, and arrows indicate voids), and there were voids near the junction of the larger SiC particles and the matrix and in the SiC segregation region. Because of the poor wettability and poor bonding strength between the SiC particles and the aluminum matrix, voids were formed at the interface. From Figure 4b, it can be seen that there were a large number of dispersed particles in the matrix of the composite material. The second phase of the dispersed particles in the matrix of the composite material was measured by EDS. The results are shown in Table 3. According to the related literatures and the XRD analysis, the AlCu and AlCuMg phases precipitated on the particle boundary and the SiC particle surface. Through further observation and analysis, it was found that the AlCu and AlMgCu phases that precipitated on the surface of the SiC particles also had the characteristics of preferred nucleation, and they tended to grow on the particle boundary and at the edges and corners of the SiC particles.

In Table 3, it can be seen that the C and Si contents near the SiC particles were higher than those of the Al matrix. The farther away from the SiC particles, the lower the content of C and Si. Figure 4d shows a line scan near the SiC particles. 

During the deposition process, the environment temperature of the composites was relatively high. SiC particles react with 7055 Al droplets after atomization. A part of the formed Si diffused into the 7055 Al matrix. At the same time, the surface of SiC dissolved to form Si and C that diffused to the matrix [37,38], forming the gradient region of Si and C shown by EDS midline scanning.

Figure 4f,h,i are elemental surface scanning maps of Si, Mg and Cu in the closed region of the SiC particle group. From these maps, as well as Figure 4e, it can be seen that there was a segregation region with a staggered distribution of Si, Mg and Cu precipitates and matrix-like phases in the closed region of the SiC particle group. EDS analysis of the element-enriched phase and matrix-like phase showed that the proportion of Si, Mg and Cu elements in the precipitated phase were 1260%, 1210% and 715% higher than that of the matrix-like phase in the annular sealing zone of the SiC particles, and 21670%, 880% and 457% higher than that of the 7055 alloy respectively.

In this study, the diffusion of SiC particles reacting with the matrix and dissolving into the matrix followed non-oriented selective free diffusion. However, the EDS surface scan results showed that the diffusion of the Si elements in the annular enclosure region followed oriented selective diffusion, (see Table 4). The diffusion direction of the Si elements in the region was from the outside to the inside of the annular SiC particle group, and finally, in the annular enclosure region. A new segregation region of Mg, Si and Cu elements formed in the center, but this phenomenon was not seen in other regions. This was due to the formation of Mg and Cu segregation zones in the matrix of this region during spray deposition, which resulted in the orientation-selective diffusion of Si elements. From Figure 4f,h–j, it can be seen that the enrichment zones of Mg, Si and Cu had obvious phase boundaries with the matrix. During the deposition process, the deposited billet was in a high temperature environment, thus the Si element was able to be fully diffused and tended to diffuse to the enrichment zones of Mg and Cu, forming a new segregation zone.

Figure 5 is a transmission electron microscopic image of the microstructure of the composite material. The precipitation of nanoparticles with a size of about 90 nm was observed in the matrix. Further observation showed that there was an obvious dislocation accumulation on one side of the nanoparticles (yellow region), while the dislocation on the other side was relatively rare (red region). This was the result of nanoparticles hindering dislocation movement.

### 3.3. Mechanical Properties

Table 5 shows the Vickers hardness of the composites. The hardness of the SiC particles and Al matrix was 10.20 HV0.05 and 8.44 HV0.05, respectively (see Table 5). From Figure 6, the hardness of the SiC particles was higher than that of the Al matrix, and it was hard and brittle. Generally, the hard particles in the material make it easy to cause a stress concentration and become the source of cracks. However, we observed that the SiC particles in the Al matrix were also able to hinder crack growth, which may be one of the reasons why SiC particles can be used as phase reinforcement, as shown in Figure 7.

## 4. Conclusions

(1) Spray-deposited SiCp/7055 Al matrix composites are mainly composed of alpha-Al, SiC, Al_2_CuMg, Al_2_Cu and MgZn_2_. The aluminum liquid particle size of materials are different, and the phenomenon of large particles extruding small particles is common in the area surrounded by SiC particles, due to the asynchronization of nucleation caused by the different cooling rates of the SiC particles and the Al matrix.

(2) C and Si on the surface of SiC particles diffuse to the Al matrix. The distribution of the two elements followed the pattern of gradient weakening around the SiC particles—i.e., a ring-shaped enrichment zone of C and Si formed around the SiC particles—and a new enrichment zone of Mg, Si and Cu formed in the closed area of the SiC particles.

(3) The AlCu and AlMgCu phases in the composites precipitated not only at the aluminum liquid particle boundaries, but also on the surface of the SiC particles, and had the characteristics of preferred nucleation. They tended to grow at the edges and corners of the SiC particles.

(4) SiC particles can hinder the propagation of tear edges and cracks. Tearing edges and cracks will continue to propagate around SiC particles, which will increase their propagation journey and delay the fracture of materials.

(5) The mechanical properties of the composites in the radial direction are better than those in the axial direction. The tensile strength and elongation after fracture were 17.4% and 8% higher, respectively. The fracture morphology of the composites was mainly composed of cleavage platforms and a few dimples, generally of the brittle fracture type.

## Figures and Tables

**Figure 1 materials-12-01299-f001:**
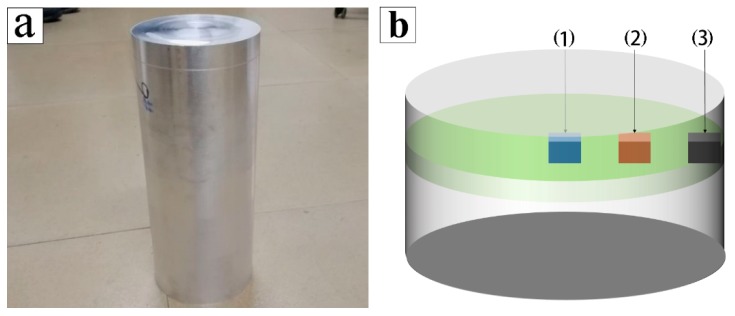
Macroscopic morphology and sampling location of spray-deposited SiC_p_/7055 aluminum composites: **a**: sample macroscopic morphology, **b**: sampling location diagram: (**1**) at the center, (**2**) at the center of the radius, (**3**) at the edge.

**Figure 2 materials-12-01299-f002:**
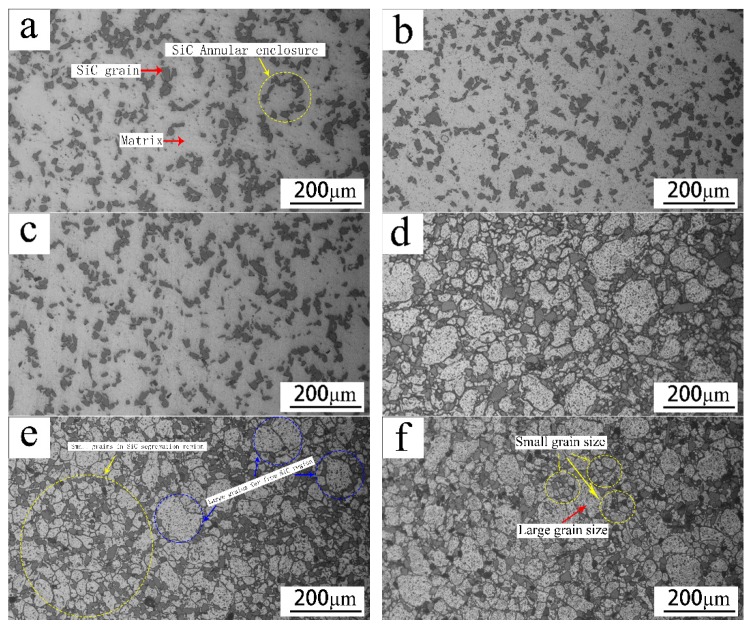
Metallographic structure of spray-deposited SiC_p_/7055Al composites at different locations: (**a**–**c**) were deposited at different locations, and (**d**–**f**) are particle structures at the corresponding locations.

**Figure 3 materials-12-01299-f003:**
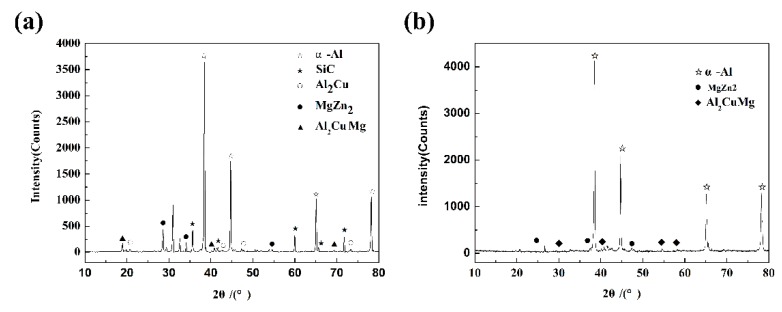
XRD analysis of spray-deposited SiC_p_/7055 Al matrix composites and 7055 Al matrix: (**a**) SiCp/7055 Al matrix composites (**b**) 7055 Al matrix.

**Figure 4 materials-12-01299-f004:**
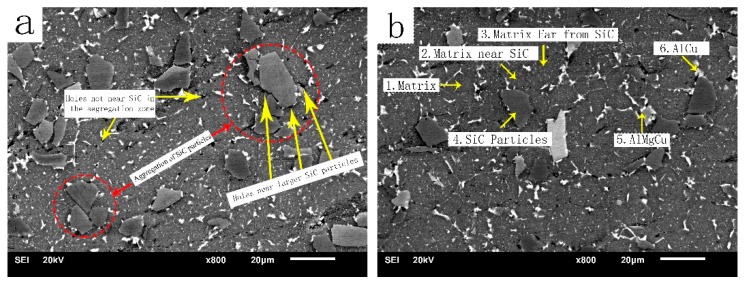
Scanning electron microscopy photos of spray-deposited SiC_p_/7055 Al matrix composites: (**a**,**b**) scanning image; (**c**) line scanning image; (**d**) line scanning distribution of C and Si elements in (**c**); (**e**) annular closed area scanning image; (**f**) enlarged area map of (**e**); (**g**–**k**) elemental distribution maps of Al, Mg, Si, Cu, C and Zn, respectively.

**Figure 5 materials-12-01299-f005:**
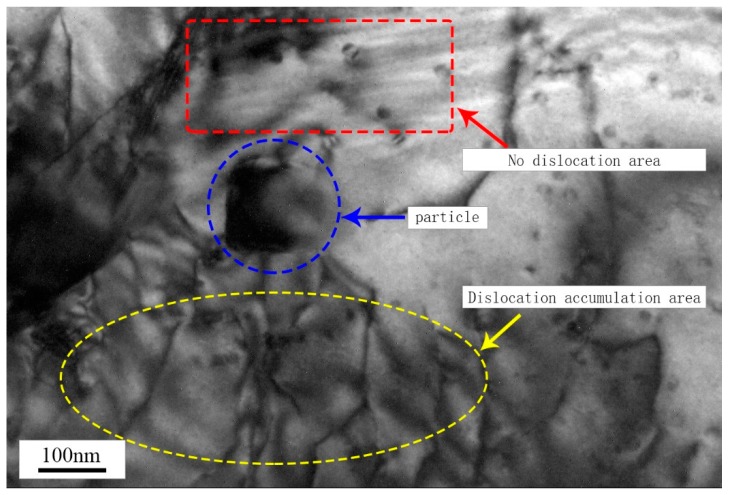
TEM photos of spray-deposited SiCp/7055 Al matrix composites.

**Figure 6 materials-12-01299-f006:**
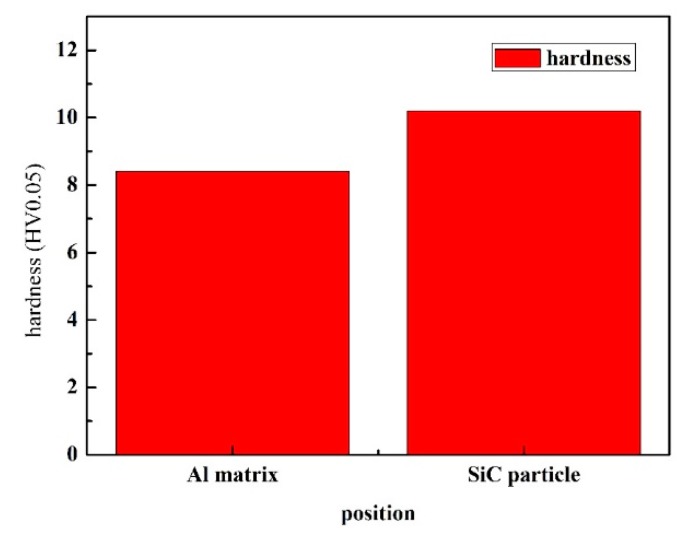
Matrix and SiC hardness tests of spray-deposited SiC_p_/7055 aluminum matrix composites.

**Figure 7 materials-12-01299-f007:**
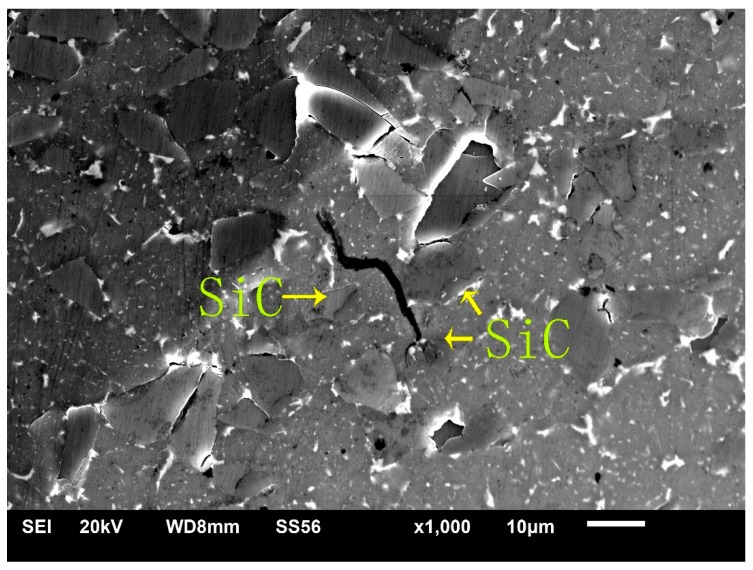
SiC particles in spray-deposited SiC_p_/7055 aluminum matrix composites prevent crack propagation.

**Table 1 materials-12-01299-t001:** Nominal composition of 7055 aluminum alloy.

Element	Si	Fe	Cu	Mn	Mg	Cr	Zn	Ti	Zr	Al
Content (%)	0.1	0.15	2.0–2.6	0.05	1.8–2.3	0.4	7.6–8.4	0.06	0.08–0.25	Other

**Table 2 materials-12-01299-t002:** Spray deposition process parameters.

Experimental Parameters	Numerical Value	Unit
Atomization temperature	750~850	°C
Nebulizer pressure	0.6~0.8	MPa
Diameter of sedimentary disk	530	mm
Matrix rotation speed	150~250	r/min
Powder-feeding pressure	0.1~0.2	MPa

**Table 3 materials-12-01299-t003:** EDS test results (wt.%).

Number	Second Phase	Al	Zn	Mg	Cu	Si	C
1	Al matrix	91.1	6.4	0.9	2.5	0	0
2	Al matrix near SiC	75.7	5.7	0	1.2	7.3	10.1
3	Al matrix far from SiC	84.6	5.5	0	1.3	0	8.6
4	SiC	0	0	0	0	67.4	32.6
5	AlMgCu phase	46.5	3.6	15	9.8	18.2	0
6	AlCu phase	59.8	3	0	36	1.2	0

**Table 4 materials-12-01299-t004:** EDS test result for the annular closed zone (wt.%).

Test Location	Si	Mg	Cu	Others
Phases of Si, Mg, Cu	21.77	19.65	12.80	Bal.
Matrix phase in the precipitation zone	1.60	1.50	1.57	Bal.
Matrix phase	0.1	1.8~2.3	2.0~2.6	Bal.

Bal. represents the mass fraction of Al and other trace elements.

**Table 5 materials-12-01299-t005:** Vickers hardness of spray-deposited SiC_p_/7055 Al matrix composites.

Position	Hardness/HV
Al matrix	8.44 HV0.05
SiC particle	10.20 HV0.05

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
