# Peer review of "Study of the Microstructure and Ring Element Segregation Zone of Spray Deposited SiCp/7055Al"

_materials, 2019, doi:10.3390/ma12081299_

Round 1
Reviewer 1 Report
Dear authors,
thank you for your contribution “Study on Microstructure and Ring Element Segregation
3 Zone of Spray Deposited SiCp/7055Al”.
I found the following issues:
Section 1:
· You did not mention main spray conditions of the spray forming process (melt flow, gas flow, GMR, superheat). Was the SiC-powder injected into the spray cone (what was the powder flow rate?)? Please add.
· Please add dimensions of the sprayed deposit – it is not clear, if the whole deposit has been used or only a small part. This may have consequences, if the microstructure of the deposit if not uniform – especially porosity. What was the porosity level in the as-sprayed condition? How was the distribution of SiC in the as-sprayed condition?
· It is not clear, how the deposit was processed to a cylinder. Machining & Extrusion? Please include this information. What was the porosity level after extrusion? Is there porosity encapsulated by SiC? This is a common problem with spray forming of MMCs.
Section 2:
· Grain size, line 88: what you see is not the grains, but the individual aluminum droplets/particles which between the SiC. Maximum diameter is around 100 µm, which is a typical particle size for spray forming. Keep in mind that the smaller ones may be a lot larger but being cut in a different plane.
· Line 118: “Due to the rapid cooling rate of spray deposition, the alloy lacks liquid phase feeding in the deposition process, and the gas may enter into the droplet under the action of high-speed atomization gas flow, thus forming voids in the matrix”
This is not clear to me. If gas enters a liquid droplet, it will usually break up. Porosity or voids in spray forming are introduced by the mechanisms “hot porosity” and “cold porosity”. Even if bubbles were introduced into the droplets, their scale would be in the range of 50% of the droplet diameter which is not the scale of your voids. Looking at your micrographs, I assume that you were on the “cold side” of the process due to the large amount of SiC particles included. To me, your statement of poor wettability if more consistent.
· Line 139: “SiC particles react with 7055Al droplets after atomization”: Did you find overspray particles which included SiC to prove this statement?
· Line: 190, hardness: it appears that the hardness of the matrix and the SiC-particles were measured separately (nanoindentation?). The SiC value is fairly low, which means that not the actual hardness was measured but the the SiC particle was pushed into the soft matrix. Such a measurement is meaningless. Instead, a compression test, tensile test or bending test should be done. Such a tests includes a volume large enough to actually measure the interaction of the soft matrix with the SiC particles.
References:
The references are missing a lot of fundamental international work which was done on the spray forming process and production of MMC. Especially the system Aluminum+SiC was investigated quite detailed. Please do a broader literature research.
Author Response
Response to Reviewer 1 Comments
Editorial reference number: materials-469792
Reviewers’ comments (materials-469792, Editor’s email on March 25, 2019):
The authors would like to sincerely thank the Reviewers and the Editor for taking the time to review our manuscript and for providing us with constructive feedbacks and excellent suggestions to improve our paper. We have addressed all the comments as best we could.
Point 1: You did not mention main spray conditions of the spray forming process (melt flow, gas flow, GMR, superheat). Was the SiC-powder injected into the spray cone (what was the powder flow rate?)? Please add.
Response 1: Many thanks for the reviewer’s comment. In line 91 of the manuscript, I added Table 2 to list the detailed parameters of spray deposition. I have listed all the parameters I know. If you need more parameters, please let me know.
Point 2: Please add dimensions of the sprayed deposit – it is not clear, if the whole deposit has been used or only a small part. This may have consequences, if the microstructure of the deposit if not uniform – especially porosity. What was the porosity level in the as-sprayed condition? How was the distribution of SiC in the as-sprayed condition?
Response 2: In line 92 of my manuscript, I added a sampling schematic to describe my sampling location, and I described the distribution of SiC particles in line 105-107. If these changes do not answer your suggestions well, please let me know how to improve.
Point 3: It is not clear, how the deposit was processed to a cylinder. Machining & Extrusion? Please include this information. What was the porosity level after extrusion? Is there porosity encapsulated by SiC? This is a common problem with spray forming of MMCs
Response 3: Thank you for your valuable request for my manuscript.In line 81-83, which is newly added to my manuscript, I describe the sample preparation and void fraction in more detail, hoping to answer your questions.
Section 2:
Point 1: Grain size, line 88: what you see is not the grains, but the individual aluminum droplets/particles which between the SiC. Maximum diameter is around 100 µm, which is a typical particle size for spray forming. Keep in mind that the smaller ones may be a lot larger but being cut in a different plane.
Response 1: Thank you for your insightful questions. I have replaced the description of grain in this article with that of aluminium droplets/particles.
Your idea that small particles may be large particles is very important. In my article, I discuss a flat surface, which is not as comprehensive as your vision. As you said, in three-dimensional space, a small part of the plane may be exposed, but in fact it is a very large particle. In my research, the part of the particle exposed on this plane is discussed, which can not reflect the nucleation and growth of the whole aluminum droplet, but the cooling nucleation of such a large aluminum droplet will certainly be affected by various conditions. On the plane I studied, its nucleation and growth law may be a small part of a large size Aluminum droplet, but it is in the process of cooling nucleation. The nucleation and growth in the plane are also independent.
Point 2: Line 118: “Due to the rapid cooling rate of spray deposition, the alloy lacks liquid phase feeding in the deposition process, and the gas may enter into the droplet under the action of high-speed atomization gas flow, thus forming voids in the matrix”
This is not clear to me. If gas enters a liquid droplet, it will usually break up. Porosity or voids in spray forming are introduced by the mechanisms “hot porosity” and “cold porosity”. Even if bubbles were introduced into the droplets, their scale would be in the range of 50% of the droplet diameter which is not the scale of your voids. Looking at your micrographs, I assume that you were on the “cold side” of the process due to the large amount of SiC particles included. To me, your statement of poor wettability if more consistent.
Response 2: Thank you for your doubts. I think you're right. I've been thinking about revising it for several days, but I didn't think of a good way to do it. So I've deleted this description from my manuscript. I hope I can solve this problem in this way.
Point 3: Line 139: “SiC particles react with 7055Al droplets after atomization”: Did you find overspray particles which included SiC to prove this statement?
Response 3: Thank you for pointing out the questions I wrote in this section. After I saw the diffusion of Si element, I tried to explain how a large number of Si was produced. Through the literature, I found that the transition layer you said was observed by predecessors, so this formula is valid, so I quoted reference 35 in the manuscript to explain it. You're right, so I changed the description and explanation of this phenomenon in the manuscript. I hope you can give me better suggestions after checking it.
Point 4: Line: 190, hardness: it appears that the hardness of the matrix and the SiC-particles were measured separately (nanoindentation?). The SiC value is fairly low, which means that not the actual hardness was measured but the the SiC particle was pushed into the soft matrix. Such a measurement is meaningless. Instead, a compression test, tensile test or bending test should be done. Such a tests includes a volume large enough to actually measure the interaction of the soft matrix with the SiC particles.
Response 4:Thank you for questioning me on this point. I have measured the Vickers hardness of SiC. As you said, this is not the true hardness of SiC particles. In this way, I want to observe the performance of SiC particles under the action of force in the matrix. As you mentioned, whether it is pushing or other, but the same force acts on SiC particles and matrix, there are different realizations. This is the purpose of my experiment, not to test the hardness of SiC particles. So I haven't changed that for the time being. If I don't explain it clearly enough, I hope you can tell me to let me improve.
References:
The references are missing a lot of fundamental international work which was done on the spray forming process and production of MMC. Especially the system Aluminum+SiC was investigated quite detailed. Please do a broader literature research.
Response:Thank you for your comprehensive analysis of my manuscript. I think your request is necessary. After reading the literature, I added a summary of jet technology and SiC particle reinforced composites in the introduction (lines 49-67). At the same time, I added these documents to the references.
Thank you very much for reviewing my manuscript in your busy schedule and giving me so many valuable opinions. These are my replies to your valuable opinions. I have done my best to improve the quality of my articles. I hope to get more advice from you. I would be very happy if I could get your affirmative reply.

Reviewer 2 Report
I would suggest to present at least two XRD patterns for the bare and reinforced materials. It will be easier to follow.
For SEM , please use the word "image" instead of photograph
Experimental part requires more details on the deposition of SiC (precursors, deposition conditions, deposition temperature, etc.). The technical parameters of XRD, SEM, EDX measurements are also missed.
Not mentioned how many times the EDX measurements were taken from each area. Are the EDX results presented from one measurement or from several ones?
Author Response
Response to Reviewer 1 Comments
Editorial reference number: materials-469792
Reviewers’ comments (materials-469792, Editor’s email on March 25, 2019):
The authors would like to sincerely thank the Reviewers and the Editor for taking the time to review our manuscript and for providing us with constructive feedbacks and excellent suggestions to improve our paper. We have addressed all the comments as best we could.
Point 1: I would suggest to present at least two XRD patterns for the bare and reinforced materials. It will be easier to follow.
Response 1: Many thanks for the reviewer’s advice.In line 145 of the manuscript, I inserted the spray deposition 7055 XRD diffraction analysis (Fig. 3b).
Point 2: For SEM , please use the word "image" instead of photograph.
Response 2: Thank you very much for pointing out such an important vocabulary error for me. I have changed all the “SEM photograph” in this manuscript to “SEM image”.
Point 3: Experimental part requires more details on the deposition of SiC (precursors, deposition conditions, deposition temperature, etc.). The technical parameters of XRD, SEM, EDX measurements are also missed.
Response 3: Many thanks for the reviewer’s comment. In line 71 of the manuscript, I added Table 2 to list the detailed parameters of spray deposition.
And I list the XRD parameters in lines 138-1140 of the manuscript, list the SEM and parameters in lines 148-150 of the manuscript.
Point 4: Not mentioned how many times the EDX measurements were taken from each area. Are the EDX results presented from one measurement or from several ones
Response 4: Thank you for your reminder. I did EDX five times for each area and then averaged it.
Many thanks for the reviewer’s comment. All spelling errors have been corrected
Thank you very much for reviewing my manuscript in your busy schedule and giving me so many valuable opinions. These are my replies to your valuable opinions. I have done my best to improve the quality of my articles. I hope to get more advice from you. I would be very happy if I could get your affirmative reply.

Round 2
Reviewer 1 Report
Dear authors,
thank you for your revision of the document